# Impact of Face-to-Face Teaching in Addition to Electronic Learning on Personal Protective Equipment Doffing Proficiency in Student Paramedics: Randomized Controlled Trial

**DOI:** 10.3390/ijerph19053077

**Published:** 2022-03-05

**Authors:** Ludivine Currat, Mélanie Suppan, Birgit Andrea Gartner, Emmanuel Daniel, Mathieu Mayoraz, Stephan Harbarth, Laurent Suppan, Loric Stuby

**Affiliations:** 1A.C.E. Genève Ambulances, Emergency Medical Services, CH-1225 Chêne-Bourg, Switzerland; l.currat@ace-ambulances.ch; 2Division of Anesthesiology, Department of Anesthesiology, Clinical Pharmacology, Intensive Care and Emergency Medicine, University of Geneva Hospitals and Faculty of Medicine, CH-1211 Geneva, Switzerland; melanie.suppan@hcuge.ch; 3Division of Emergency Medicine, Department of Anesthesiology, Clinical Pharmacology, Intensive Care and Emergency Medicine, University of Geneva Hospitals and Faculty of Medicine, CH-1211 Geneva, Switzerland; birgit.gartner@hcuge.ch (B.A.G.); laurent.suppan@hcuge.ch (L.S.); 4Infection Control Program and WHO Collaborating Centre on Patient Safety, University of Geneva Hospitals and Faculty of Medicine, CH-1211 Geneva, Switzerland; emmanuel.daniel@hcuge.ch (E.D.); stephan.harbarth@hcuge.ch (S.H.); 5MEDI—Center for Medical Education, College of Higher Education in Ambulance Care, CH-3014 Bern, Switzerland; mathieu.mayoraz@medi.ch; 6Genève TEAM Ambulances, Emergency Medical Services, CH-1201 Geneva, Switzerland

**Keywords:** personal protective equipment, electronic learning, prehospital, student paramedics, infection prevention, face-to-face learning, randomized controlled trial, Peyton’s approach, blended learning

## Abstract

Personal protective equipment doffing is a complex procedure that needs to be adequately performed to prevent health care worker contamination. During the COVID-19 pandemic, junior health care workers and students of different health care professions who had not been trained to carry out such procedures were often called upon to take care of infected patients. To limit direct contact, distance teaching interventions were used, but different trials found that their impact was rather limited. We therefore designed and carried out a randomized controlled trial assessing the impact of adding a face-to-face intervention using Peyton’s four-step approach to a gamified e-learning module. Sixty-five student paramedics participated in this study. The proportion of doffing sequences correctly performed was higher in the blended learning group (33.3% (95%CI 18.0 to 51.8) versus 9.7% (95%CI 2.0 to 25.8), *p* = 0.03). Moreover, knowledge and skill retention four to eight weeks after the teaching intervention were also higher in this group. Even though this study supports the use of a blended learning approach to teach doffing sequences, the low number of student paramedics able to adequately perform this procedure supports the need for iterative training sessions. Further studies should determine how often such sessions should be carried out.

## 1. Introduction

### 1.1. Background and Importance

The COVID-19 pandemic has highlighted the key role of personal protective equipment (PPE) and revealed that many HCWs lack both knowledge and training regarding this equipment [1,2]. Efficient and adequate PPE use is, however, of paramount importance to avoid contaminating both patients and HCWs [3,4]. While correctly donning PPE is of critical importance, many studies have shown that contamination usually occurs during the doffing phase [5,6,7,8,9,10]. In addition, donning and doffing PPE in non-dedicated spaces with time constraints increase the contamination risk even further [11,12].

Training HCWs is an efficient way of decreasing self-contamination during PPE doffing [13,14,15,16,17]. A recent systematic review concluded to the superiority of face-to-face training over passive training only (i.e., text documents or video) regarding PPE doffing procedure compliance [18]. However, the evidence retrieved was considered to be of rather low quality; thus, the authors highlighted the need of further clinical trials for comparing training methods. 

A gamified e-learning module tailored to the needs of prehospital providers was previously developed [19]. The impact of this intervention was however limited [20,21]. While distance learning was strongly promoted during the first wave of the COVID-19 pandemic [22,23], complex skills undoubtedly require a blended learning approach, combining workshops and other teaching interventions [24,25,26]. Such workshops could rely on Peyton’s four-step educational approach [27], which has been shown to be more efficient at teaching procedural skills to HCWs than a standard teaching approach [28]. Our hypothesis was that using a blended learning strategy by adding a workshop using Peyton’s approach to an interactive, gamified e-learning module would increase knowledge and skill acquisition and retention regarding PPE doffing procedures in student paramedics. Adhering to Peyton’s approach was thought to be of particular interest since this method is also used during continuous training sessions followed by paramedics in Switzerland [29,30].

### 1.2. Objectives

The goal of this randomized clinical trial (RCT) was to determine whether adding a face-to-face teaching intervention (following Peyton’s approach) to a gamified e-learning module could improve correct doffing sequences’ skills and knowledge and retention in student paramedics.

## 2. Materials and Methods

### 2.1. Study Design and Setting

This was a parallel-group, randomized, triple-blind (participants, instructors, and outcome assessors) controlled superiority trial prospectively registered (International Registered Report Identifier PRR1-10.2196/26927) (Figure 1). The protocol has been previously published [31]. It is reported according to the Consolidated Standards of Reporting Trials (CONSORT)-EHEALTH checklist [32] and includes relevant elements from the Checklist for Reporting Results of Internet E-Surveys (CHERRIES) since online questionnaires were used in this study [33]. The regional ethics committee delivered a declaration of non-objection as to conducting this RCT (Req-2020-01340).

All first-year students (*n* = 65) from the Colleges of Higher Education in Ambulance Care located in Bern and Geneva, Switzerland, were invited to participate in this study. To allow the first-year German-speaking students to participate in the study, the study material including the e-learning module was translated in German. There were no exclusion criteria. No incentive was provided.

Instructors, recruited for study purposes, were informed that the objective was to teach the non-contaminating doffing of PPE to first-year students during two training sessions. They were not aware of the study design, and were therefore blinded to the existence of two different training paths. The instructors received a detailed PPE doffing procedure and a summary sheet of the Peyton’s approach steps as a reminder during the sessions. They were previously trained by investigators about the doffing procedure and Peyton’s pedagogical approach. The instructor:learner ratio was planned to range from 1:1 to 1:3 as such ratios have been shown to be particularly effective [28].

### 2.2. Online Platform

An online platform [34] running under the Joomla! 3.9 content management system (Open Source Matters) was developed by LSu for the study’s purpose and thoroughly tested by 3 other authors. 

### 2.3. Randomization and Concealment of Allocation

An investigator (MS), who did not know the participants and had no contact with them, created unique study accounts, which were then randomized into two groups according to a computer-generated list [35] with a 1:1 allocation ratio and stratification by school and language. Opaque, sealed envelopes containing individual login information were created and transmitted to local investigators. Given the utter lack of risk for the participants, there was no unblinding procedure, no data monitoring, and no interim analysis. Participants were randomly divided between the instructors by an investigator (LSt or LC) using an online team generator [36].

### 2.4. Enrolment and Consent

Prior to the beginning of the study, an email containing information about the study was sent to the participants. Prior to logging in, the learning objectives and a data security statement were displayed on the website home screen. Consent was obtained electronically.

### 2.5. Study Sequence

After clicking the start button, a first questionnaire designed to collect demographic data and assess learning differences according to the VARK (visual, aural, reading/writing, or kinesthetic) modalities was displayed. Both groups were then invited to follow an interactive, gamified e-learning module [19].

After completing the module, participants belonging to the control group were asked to don the following PPE: protective glasses, N95 respirator mask, coverall with hood, and gloves. They were then asked to perform individually the doffing sequence, which was recorded on video. After completing this procedure, participants were asked to return to the online platform to electronically rebuild the doffing sequence.

Instead of immediately donning and doffing PPE after completing the e-learning module, participants in the experimental group were randomly assigned to one of the instructors to follow face-to-face learning according to Peyton’s 4-step approach. After this workshop, these participants resumed the same path as their peers from the control group by moving on to the recording of the doffing sequence on video and were then asked to rebuild the doffing sequence on the online platform.

Four to eight weeks after this initial intervention, depending on the schools’ schedules, participants were invited to a second session. Both groups first repeated the video recording of the PPE doffing sequence, then logged into the platform to electronically rebuild the doffing sequence. The experimental group was then considered as having completed the study path, while the control group attended a face-to-face learning session to ensure that all participants had ultimately been given the same level of training regardless of their initial allocation.

### 2.6. Gamified E-Learning Module

The development of the infection prevention and control (IPC) gamified e-learning module used in the present study has already been described [19]. Briefly, this module was developed following the theory-driven approach of the SERES framework [37,38]. It includes several learning objectives and provides general knowledge regarding SARS-CoV-2 (definition, incubation time, transmission routes, and symptoms), PPE items, and donning and doffing sequences. Gamification mechanisms were used to facilitate knowledge acquisition regarding donning and doffing sequences. This gamified module was also used to describe and detail the concept of “contaminated” and “noncontaminated” zones, thus separating the doffing procedure in two specific phases. The specific doffing steps occurring in each zone are detailed in Appendix A.

### 2.7. Face-to-Face Learning

Face-to-face teaching was based on Peyton’s approach [27] and followed these steps:(1)The instructor performed a complete doffing sequence without giving any comments;(2)The instructor performed a doffing sequence accompanied by step-by-step explanations (description of key points);(3)Learners were asked to guide the instructor through the doffing sequence, step by step;(4)Learners were asked to perform the complete doffing sequence before receiving individualized feedback. Each participant performed this step only once.

### 2.8. Primary Outcome

The primary outcome was the proportion of doffing sequences correctly performed after knowledge acquisition during the first study session. The adequacy of the procedure was individually assessed by two investigators (one of whom is an IPC specialist and the other an emergency medicine physician). These investigators, who were blinded as to group allocation, viewed the videos and completed a checklist (Appendix A). In case of disagreement, the study protocol stated that a consensus would be reached by discussion. However, due to logistic reasons, disagreements were solved by asking a third investigator (MS) to individually review the recordings. This investigator was not provided with the ratings given by the other investigators. Her assessments allowed us to address all disagreements as all the ratings were binary (either correct or incorrect).

### 2.9. Secondary Outcomes

Nine secondary outcomes were assessed. Seven of them were prespecified in the study protocol: time required to teach the technique, time required to perform the doffing procedure, learner satisfaction, proportion of correct computer sequences, confidence in using PPE, and knowledge and skill retention. The last two were decided during study sessions and before the constitution of the database: proportion of correct hand disinfection, and number of errors (procedure deviations and/or contaminations) during the procedure. 

Analyzing the proportion of correct hand disinfection was decided as some investigators quickly noticed that, during the study sessions, some of the hand disinfection procedures were incorrect. As this action was not specifically taught in either group, it was decided to consider that the doffing procedure was correct if the hand disinfection procedure was performed at the correct time point, even if it was poorly executed. The entire PPE doffing procedure, which was validated by IPC experts, is detailed in Appendix A. However, as hand disinfection is a critical IPC procedure, it was decided to assess it separately. Adequate friction time and proper disinfection of all hand’s areas were assessed subjectively.

The number of errors was added as secondary outcome because it allows a more precise assessment of the overall performance than a binary outcome. Furthermore, the risk of contamination is probably not binary and would be expected to increase as the number of errors increases.

### 2.10. Blinded Data Collection and Assessment

Some outcomes were recorded electronically allowing their assessment to be independent from subjective human assessment. For the other outcomes, assessors were blinded to participants’ allocation. In case of disagreements, we chose to ask MS to review the recordings in order to reach a consensus. 

### 2.11. Data Availability

All investigators were able to access the curated and coded data set. The database is available as Appendix A. The video recordings were used only for the purpose of this study and were destroyed after analysis.

### 2.12. Sample Size

In two previous studies, no participant was able to electronically rebuild the correct doffing sequence after following the gamified e-learning module used in the present study [20,21]. However, the practical reality (skill) could be dissociated from the theoretical responses (knowledge) collected on the online platform, which made it necessary to set up a control group in this study. 

It was calculated that 46 participants would be needed to have a 90% chance of detecting, at the 5% significance level, an increase in the primary outcome from 10% in the control group to 50% in the experimental group; additional participants were accepted as the training was part of their curriculum.

### 2.13. Statistical Analysis

Data analysis was performed using Stata 15.1 (StataCorp. 2017. Stata Statistical Software: Release 15. StataCorp LLC, College Station, TX, USA). Due to the small sample size, only non-parametric tests were used. Fisher’s exact test was used for dichotomous variables and the Mann–Whitney U test for continuous variables. The computerized doffing sequence accuracy was analyzed as a whole, and according to the respective doffing zones (contaminated and noncontaminated zones). The Likert scales were described graphically, and statistical comparison performed using Fisher’s exact test. Conversely to what was stated in the statistical analysis plan of the protocol, they were not dichotomized for statistical analysis (specified before statistical analysis, due to important loss of information). The results were described as a percentage with 95% CI for the proportions and according to the median (Q1;Q3) for the continuous variables. A *p* value < 0.05 was considered significant. A prespecified subgroup analysis by working status (actively working in an ambulance service or not) was carried out as an increased rate in adequate choice of PPE has been shown in this subgroup [20]. Working status was assessed through the first questionnaire. There were no missing data. 

Two post hoc analyses were decided: the estimation of the correlation between the time needed to perform the procedure and the number of errors made, and the search for an association between the VARK scores and the performance and experience (satisfaction and confidence) of the participants. For the first, after graphical description, the Pearson correlation coefficient was used. For the second, the association was sought using logistic and linear regression models with the VARK scores as predictive and adjustment variables and the performance (correct/incorrect sequence, and number of errors, respectively).

### 2.14. Protocol Deviations

The protocol deviations were the following: addition of two secondary outcomes (hand disinfection and number of errors in the doffing procedure); way to reach consensus slightly modified; unblinded statistical analysis due to sending of an unblinded data set to statistical analyst (all data had however been entirely acquired); non-dichotomization of Likert scales (decided before statistical analysis due to loss of information); use of a teacher rather than third-year students for the face-to-face teaching due to headmaster’s decision leading to the addition of a sensitivity analysis of primary outcome; face-to-face teaching ratio higher than 1:3 once, due to availability of only one German-speaking instructor during the session.

## 3. Results

Sixty-five participants were enrolled in the study. One was excluded after missing the first study session (Figure 2). Their characteristics are detailed in Table 1. All sessions but one (17/18) were conducted with an instructor:learner ratio of 1:1 to 1:3, with one exception of 1:4.

The proportion of doffing sequences correctly performed was higher in the group who followed the blended learning compared to the one who followed only the e-learning (33.3% (95%CI 18.0 to 51.8) versus 9.7% (95%CI 2.0 to 25.8), *p* = 0.03). The pre-specified subgroup analysis did not change the direction, nor the magnitude of effect (Table 2).

For the face-to-face teaching, the additional median (Q1;Q3) time required was 22 (19;25) minutes. There was no significant difference, either in the time required to perform the doffing procedure, nor in the knowledge at acquisition (independently of the zones) (Table 3). Knowledge retention and doffing skills in the contaminated zone were significantly higher in the blended learning group (Table 3).

The number of errors was significantly lower in the experimental group (Figure 3, Table 4). A weak correlation (first session, r = −0.31, *p* = 0.012, second session, r = −0.33, *p* = 0.007) was found between the time needed to perform the procedure and the number of errors made (Figure 4). There was a 0.74 error per minute decrease (95%CI 0.17 to 1.32, *p* = 0.012) during the first sessions and a 1.41 error per minute decrease (95%CI 0.40 to 2.42, *p* = 0.007) during the second sessions. The predictivity of this model is however very low (r squared = 0.1 for both sessions).

A common error concerned hand disinfection with 45.3% of overall participants performing it incorrectly during the first session, and 50.0% at the second session, with less errors in the experimental group (not statistically different) (Table 4).

Satisfaction (Figure 5) and confidence in using PPE (Figure 6) were significantly different between groups (*p* = 0.016 and *p* < 0.001, respectively).

By analyzing the primary outcome stratified according to language/type of instructor, there was no change in the direction of the effect, which was, however, only statistically significant in the French-speaking subgroup despite its smaller sample size (Table 5).

None of the VARK scores were significantly associated with the performance (regardless of binary or continuous outcome).

## 4. Discussion

### 4.1. Main Considerations

Adding a face-to-face teaching intervention following Peyton’s approach to a gamified e-learning module significantly improved PPE doffing skill acquisition in student paramedics. Prior studies have shown that adding training modalities to learning paths increases HCWs compliance and decreases contamination during PPE doffing procedures [17,39,40,41], can help decrease both the time required to doff PPE and the number of errors [42], and can allow participants to perform complex procedures more efficiently [25,26]. In our study, the superiority of the blended learning approach could be due either to the face-to-face training itself or to the addition of a second training modality. However, there was no difference in the ability of adequately rebuilding the correct PPE doffing sequence on the web platform. This supports the theory that adequately performing a complex procedure such as PPE doffing not only depends on knowledge acquisition but also relies on practical aspects. We therefore believe that using blended learning approaches, including at least a practical skill station, is necessary to teach complex procedures such as PPE doffing. This is further supported by the fact that more students felt “very confident” after following the blended learning approach.

The proportion of student paramedics adequately performing the PPE doffing procedure was lower during the follow-up sessions. However, skill retention was still higher in student paramedics who had followed both interventions in comparison with those who had only followed the e-learning module. Once again, there was no significant difference between groups in the ability of adequately rebuilding the full PPE doffing sequence on the web platform. 

During both sessions, the time required to perform the doffing procedure was similar between groups, with shorter times associated with a slightly increased number of errors. Importantly, participants who had only followed the e-learning module made twice as many errors than those who had followed the blended learning approach. Errors during the doffing procedure can lead to self-contamination and increase HCW’s risk to contract or transmit diseases [9,43,44,45]. The more errors HCWs make, the higher the risk [43]. However, not every deviation leads to contamination [43,46], and the risk should therefore be considered according to the kind of error made by the HCW [47]. Since this outcome was not included in the present study, possible contaminations were assessed subjectively without using marking agents (phosphorescence or inactivated virus for instance) [10,43].

Almost one paramedic student out of two did not adequately disinfect hands. Since this procedure is considered as a basic skill, it was neither taught nor reminded in both study arms. Hand disinfection is nevertheless a critically important procedure, and emphasis will need to be put on the importance of adequately performing hand disinfection [48]. Given these unexpectedly low results, particularly in the context of the current pandemic, reminders should therefore be included in all future IPC teaching interventions [49,50] owing to the contamination risk associated with improper hand disinfection [46,51].

In line with prior studies [9,52,53], the participants who were not actively working in an ambulance service had higher skill acquisition and retention scores. Some authors suggest that complex procedures such as PPE doffing should be taught to HCWs before they acquire inadequate practices [13]. However, this finding could also be due to confounding biases regarding several factors. Indeed, all paramedic students are required to work actively in an Emergency Medical Service (EMS) once they are admitted to one of the two schools involved in this study (Bern’s school). Since this school is the only one including German-speaking participants who were taught by a full-fledged teacher rather than by third-year students, the differences seen in our study could be, in part if not entirely, due to these differences.

### 4.2. Limitations and Strengths

This study has several limitations other than the protocol deviations already listed above. First, even though we strived to increase the number of participants by carrying out a multicenter study and translated the previously described e-learning module into German for this purpose, our sample size was still small. This small sample size precluded any strata analysis, and biases linked to the multicenter and multilingual aspects of our study cannot be ruled out. In addition, the planned sample size was not adequate and the post hoc analysis yields a power of 54%. Indeed, we had correctly anticipated that some student paramedics would adequately perform the PPE doffing procedure despite the fact that prior results showed that almost all participants were unable to correctly rebuild the doffing procedure on a web platform. While this supports the theory that practical reality can be dissociated from theoretical responses, we incorrectly estimated the potential increase in PPE doffing skill proficiency after following a blended learning approach. Second, even though our goal was to use only third-year student paramedics as instructors to limit teaching biases during the face-to-face sessions, we had no choice but to use a full-fledged teacher for the subgroup of German-speaking participants. Therefore, the absence of a statistically significant difference between groups in the subset of German-speaking participants could be ascribed, at least in part, to this deviation from our original protocol, even though all instructors, regardless of their professional status, received the same training regarding Peyton’s approach. Third, we did not foresee that student paramedics might lack skills regarding hand disinfection. Even though this outcome was added after the first recordings had been assessed, there should be little bias as all assessors were blinded as to group allocation.

However, this study also has strengths. First, the robust, stratified randomization method allowed us to obtain well-balanced groups despite the limited sample size. Second, the electronic acquisition of data related to secondary outcomes also helped limit potential assessment biases. Another important strength is the utter lack of dropouts between the initial study session and the follow-up sessions. Finally, this is one of the few studies reporting the impact of adding a face-to-face teaching intervention to a gamified e-learning module.

### 4.3. Perspectives and Practical Implications

First of all, because of the differences found between theoretical responses and actual skills, future studies exploring this domain should include practical skill assessments, even if no skill station is included in the learning path. In addition, studies involving actively working HCWs should consider acquiring video recordings during actual interventions. Indeed, the number of errors in the PPE doffing procedure might actually be higher in the field because of the Hawthorne effect [54,55,56], narrow spaces, and time constraints. 

Chemical, biological, radiological, and nuclear threats are globally increasing but most health care workers (HCWs) are insufficiently trained to face such situations [57,58]. Given the unsatisfactory skill acquisition and the even poorer retention of PPE doffing procedural skills in student paramedics even after following a blended learning approach, further studies are needed to understand how the proficiency of prehospital providers can be enhanced regarding this critical procedure. While this could potentially be achieved by adding further training modalities, it is also possible that the only way of ensuring adequate skill acquisition and retention is through the provision of regular refresher sessions. The impact of regular refresher sessions has been studied regarding basic life-support skills [59,60], and future studies could be designed to assess the impact of similar sessions regarding PPE doffing skills.

Another important element that future studies should consider is the systematic assessment of computer literacy [61,62]. Indeed, even though we considered that computer literacy should be high in our population of Swiss student paramedics since the vast majority of them are young people living in a high-income country [63], the inconsistency between their actual skills and their ability to adequately rebuild the doffing sequence on a computer platform in prior studies [20] is intriguing. Therefore, we recommend that future research in this domain should always assess computer literacy in any population studied and avoid being prejudiced by potential misconceptions regarding specific populations.

Finally, iterative assessments of PPE doffing skills should be performed and PPE doffing should, whenever possible, be systematically executed under the supervision of proficient and regularly IPC-trained personnel to avoid contamination [64].

## 5. Conclusions

In this study, adding face-to-face training to a gamified e-learning module increased PPE doffing proficiency among first year student paramedics and enhanced skill retention. The proportion of participants able to adequately perform the procedure was, however, rather low, and PPE doffing procedures remain complex to learn, retain, and perform. Future studies are needed to determine whether additional training modalities can enhance skill acquisition and retention, or if regular refresher sessions are the only way of ensuring PPE doffing skills proficiency.

## Figures and Tables

**Figure 1 ijerph-19-03077-f001:**
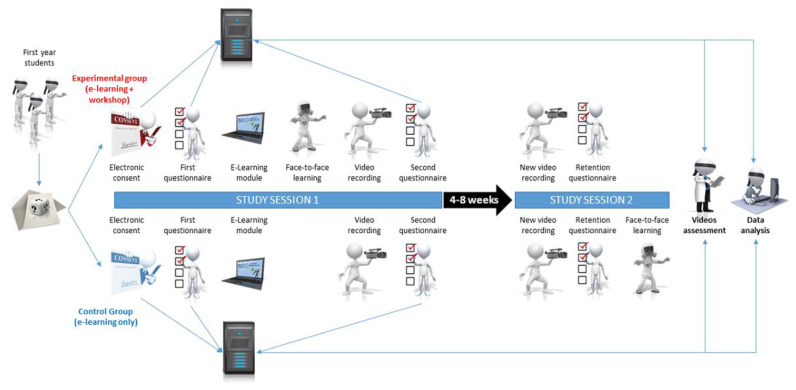
Study design, adapted from Stuby et al., 2021 [31].

**Figure 2 ijerph-19-03077-f002:**
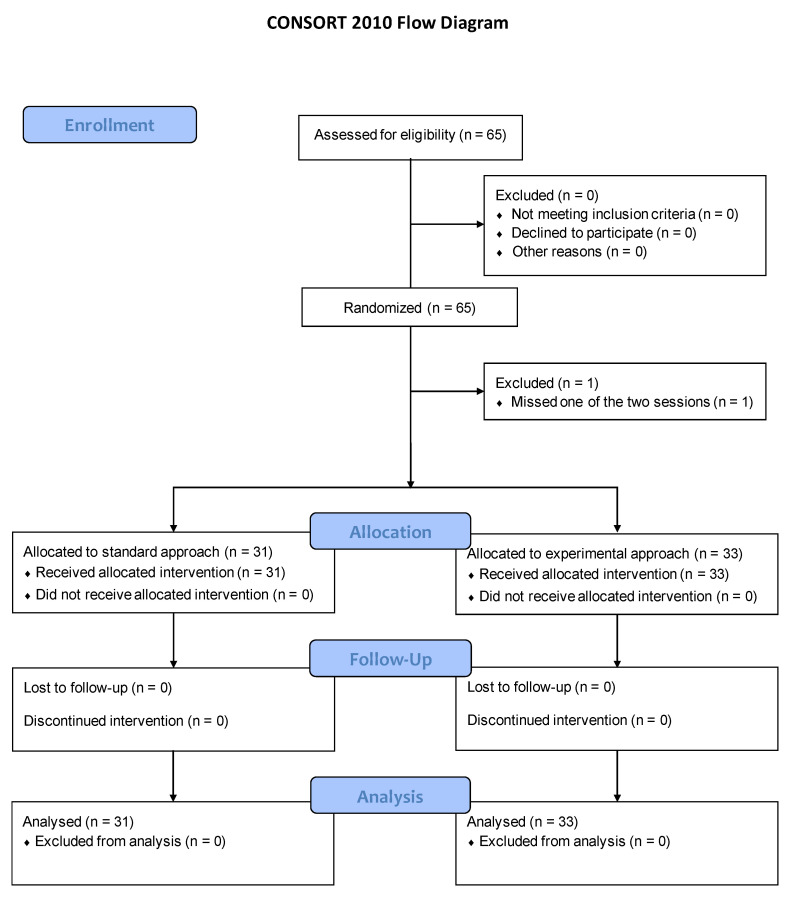
Study flowchart.

**Figure 3 ijerph-19-03077-f003:**
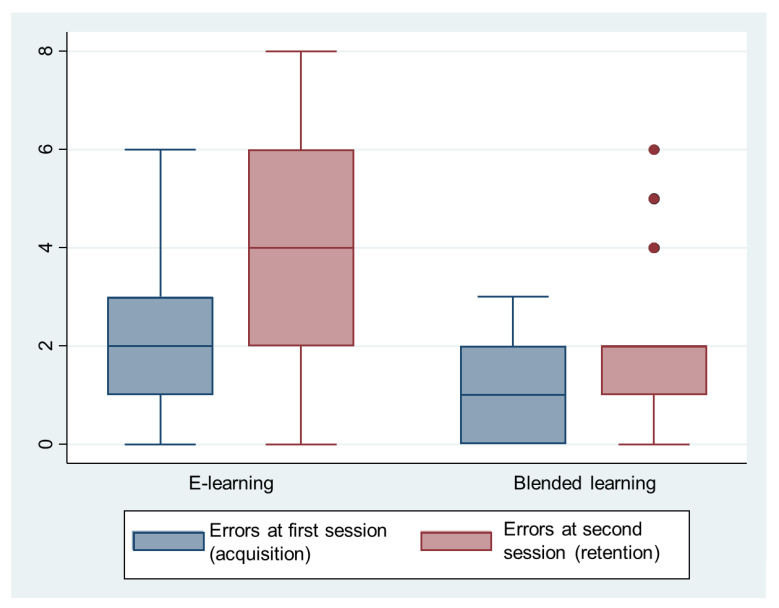
Number of errors by session.

**Figure 4 ijerph-19-03077-f004:**
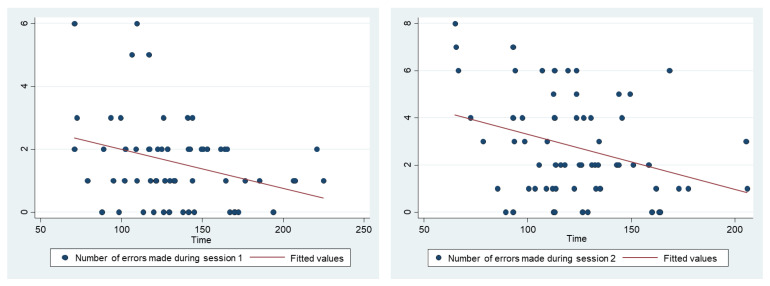
Correlation between time required to perform the doffing procedure and number of errors made.

**Figure 5 ijerph-19-03077-f005:**
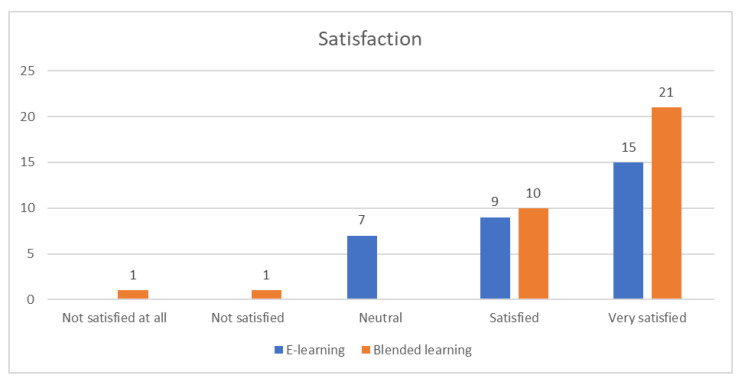
Participant satisfaction.

**Figure 6 ijerph-19-03077-f006:**
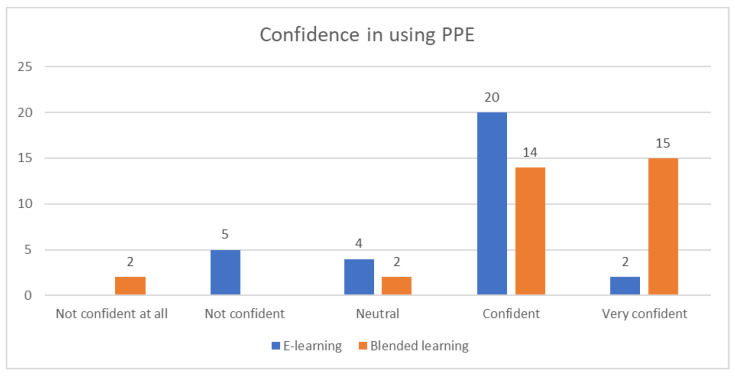
Participant confidence in using personal protective equipment.

**Table 1 ijerph-19-03077-t001:** Participants’ characteristics.

	E-Learning (n = 31)	Blended Learning (n = 33)
Age, in years, median (Q1;Q3)	24 (22;26)	27 (23;28)
Gender, n (%)		
Male	13 (41.9)	13 (39.4)
Female	18 (58.1)	20 (60.6)
Other	0 (0.0)	0 (0.0)
Location, n (%)		
Geneva	8 (25.8)	10 (30.3)
Bern (French-speaking)	5 (16.1)	5 (15.2)
Bern (German-speaking)	18 (58.1)	18 (54.6)
Actively working in an ambulance service, n (%)	24 (77.4)	24 (72.7)
Canton of practice of those currently working, n (%)		
Aargau	4 (16.7)	1 (4.2)
Basel	0 (0.0)	3 (12.5)
Bern	13 (54.2)	12 (50.0)
Fribourg	0 (0.0)	2 (8.3)
Geneva	1 (4.2)	0 (0.0)
Neuchâtel	1 (4.2)	1 (4.2)
Solothurn	1 (4.2)	2 (8.3)
Vaud	2 (8.3)	2 (8.3)
Valais	2 (8.3)	1 (4.2)
VARK scores, median (Q1;Q3)		
visual	7 (5;9)	6 (4;7)
aural	8 (5;11)	8 (6;11)
read	5 (3;6)	5 (3;7)
kinesthetic	9 (7;11)	9 (8;10)

Total may be over 100% due to rounding.

**Table 2 ijerph-19-03077-t002:** Subgroup analysis of primary outcome by working status.

	E-Learning (n = 31)	Blended Learning (n = 33)	*p*-Value
Correct sequence among participants actively working, % (95%CI)	8.3% (1.0 to 27.0)	29.2% (12.6 to 51.1)	0.14
Correct sequence among participants not actively working, % (95%CI)	14.3% (0.3 to 57.9)	44.4% (13.7 to 78.8)	0.31

**Table 3 ijerph-19-03077-t003:** Secondary outcomes.

	E-Learning (n = 31)	Blended Learning (n = 33)	*p*-Value
Time required to perform the doffing procedure at first session, in seconds, median (Q1;Q3)	133 (107;151)	129 (118;164)	0.59
Time required to perform the doffing procedure remotely, in seconds, median (Q1;Q3)	113 (93;135)	124 (113;144)	0.08
Correct computerized sequence at first session in contaminated zone (knowledge at acquisition), % (95%CI)	80.6% (62.5 to 92.5)	90.9% (75.0 to 98.1)	0.30
Correct computerized sequence at first session in non-contaminated zone (knowledge at acquisition), % (95%CI)	77.4% (58.9 to 90.4)	72.7% (54.7 to 86.7)	0.78
Correct computerized full sequence at first session (knowledge at acquisition), % (95%CI)	64.5% (45.4 to 80.8)	66.7% (48.2 to 82.0)	1.00
Correct computerized sequence remotely in contaminated zone at second session (knowledge retention), % (95%CI)	38.7% (21.8 to 57.8)	66.7% (48.2 to 82.0)	0.04
Correct computerized sequence at second session in non-contaminated zone (knowledge retention), % (95%CI)	64.5% (45.4 to 80.8)	75.7% (57.7 to 88.9)	0.42
Correct computerized full sequence at second session (knowledge retention), % (95%CI)	35.5% (19.2 to 54.6)	48.5% (30.8 to 66.4)	0.32
Doffing sequences correctly performed remotely at second session (skill retention), % (95%CI)	3.2% (0.1 to 16.8)	24.2% (11.1 to 42.3)	0.03

**Table 4 ijerph-19-03077-t004:** Number of errors by session and hand disinfection.

	E-Learning (n = 31)	Blended Learning (n = 33)	*p*-Value
Number of errors in the procedure at acquisition, median (Q1;Q3)	2 (1;3)	1 (0;2)	<0.001
Number of errors in the procedure at retention, median (Q1;Q3)	4 (2;6)	2 (1;2)	<0.001
Correct hand disinfection at acquisition, % (95%CI)	45.2% (27.3 to 64.0)	63.6% (45.1 to 79.6)	0.21
Correct hand disinfection at retention, % (95%CI)	41.9% (24.5 to 60.9)	57.6% (39.2 to 74.5)	0.32

**Table 5 ijerph-19-03077-t005:** Sensitivity analysis of primary outcome (correct doffing procedure) depending on language and type of instructor.

	E-Learning (n = 31)	Blended Learning (n = 33)	*p*-Value
French-speaking participants with third-year student as instructor (n = 28), % (95%CI)	7.7% (0.2 to 36.0)	46.7% (21.3 to 73.4)	0.04
German-speaking participants with teacher as instructor (n = 36), % (95%CI)	11.1% (1.4 to 34.7)	22.2% (6.4 to 47.6)	0.66

## Data Availability

The data presented in this study are available as Appendix A.

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
