# Peer review of "Impact of Face-to-Face Teaching in Addition to Electronic Learning on Personal Protective Equipment Doffing Proficiency in Student Paramedics: Randomized Controlled Trial"

_ijerph, 2022, doi:10.3390/ijerph19053077_

Round 1
Reviewer 1 Report
The study conducted an RCT to assess the impact of adding a face2face intervention of e-learning module of PPE doffing proficiency of students. My comments are listed as follows:
- If stratified by school and language, why not perform by strata analysis or use regression to consider the strata-group interaction? Especially there is a significant confounding factor with different instructors for different language. Also, for such a multicenter study, data from each center is a cluster and in your analysis should consider adjusting for the correlated data within each cluster.
- why use the proportion of correct performance as the primary outcome? why not use the number of errors since "it provides a more precise assessment of overall performance than a binary variable"?
- the subgroup of working status is determined before or after randomization/sampling?
- not triple blind, should be double-blinded because statistician is not blinded
- subgroup analysis did not report the p-values
- There are statistical tests to come Likert scale numeric variable instead of just showing the graphical comparison
- Your sample size calculation in the protocol power analysis is correct in theory but did not consider many factors, so can you do an ad-hoc power analysis under the observed results from the current study?
Reviewer 2 Report
The manuscript is well-written and all areas are addressed regarding the gamification.
The manuscript is easy to follow with minor grammatical errors.
My only concern is that there is mention in the methods of the article that hand hygiene is important and would not be focused on. I suggest to be more clear on when the hand hygiene indications or missed opportunities are. They are not explained in the methods, but yet there is mention about what is deemed incorrect or a missed step. Authors and IPs may be aware, but this is not clear to the general public. What is the correct timepoint?
Analyzing the proportion of correct hand disinfection was decided as some investigators quickly noticed, during the study sessions, that some of the hand disinfection procedures were incorrect. As this action was not specifically taught in either group, it was decided to consider that the doffing procedure was correct if the hand disinfection proce dure was performed at the correct timepoint even if it was poorly executed.
Round 2
Reviewer 1 Report
The authors have addressed most of my comments. I just have one more comment:
Please rephrase as "In addition, the planned sample size was not adequate and the post-hoc analysis yields ...." As I have mentioned in my previous review report, you cannot say the initial power analysis is incorrect, it is limited due to less informative prior data, not methodology.
Also, usually, power analysis reports given the fixed significance level, alpha=0.05, and the effect size calculated from the data, what is the estimated power under the current planned sample size. How come you have an "actual alpha of 0.02”??? Please double check the ad-hoc power analysis.
Author Response
Please see the attachment

This manuscript is a resubmission of an earlier submission. The following is a list of the peer review reports and author responses from that submission.